# Association between Peripheral Inflammatory Cytokines and Cognitive Function in Patients with First-Episode Schizophrenia

**DOI:** 10.3390/jpm12071137

**Published:** 2022-07-14

**Authors:** Seon-Hwa Baek, Honey Kim, Ju-Wan Kim, Seunghyong Ryu, Ju-Yeon Lee, Jae-Min Kim, Il-Seon Shin, Sung-Wan Kim

**Affiliations:** 1Department of Psychiatry, Chonnam National University Medical School, Gwangju 61419, Korea; psy20190301@gmail.com (S.-H.B.); hngo123@naver.com (H.K.); tarot383@naver.com (J.-W.K.); seunghyongryu@gmail.com (S.R.); dabium@hanmail.net (J.-Y.L.); jmkim@chonnam.ac.kr (J.-M.K.); isshin@chonnam.ac.kr (I.-S.S.); 2Mindlink, Gwangju Bukgu Mental Health Center, Gwangju 61220, Korea

**Keywords:** cognitive impairment, cytokines, first-episode schizophrenia, inflammation, psychosis

## Abstract

In this study, we investigated the impact of inflammatory cytokines on the cognitive performance of patients with schizophrenia. The included patients met the criteria for schizophrenia spectrum disorder and were aged between 15 and 40 years, with a duration of illness ≤1 year. Plasma tumor necrosis factor (TNF)-α; interferon-γ; and interleukin (IL)-1β, IL-6, IL-8, IL-10, and IL-12 levels were measured. A computerized neurocognitive battery, measures for social cognitive function, and clinical measures were administered. A total of 174 patients with first-episode psychosis were enrolled. The TNF-α level was negatively correlated with scores on the digit span, verbal learning, and Wisconsin card sorting tests, and the number of correct responses on the continuous performance test (CR-CPT), whereas a positive correlation was detected with the trail making test (TMT)-B time. The interferon-γ level was negatively correlated with performance on the false belief and visual learning tests. The IL-1β level was positively correlated with the TMT-A time and CPT reaction time, whereas it was negatively correlated with the CR-CPT and performance on the visual learning and social cognitive tests. The IL-12 level was negatively correlated with the CR-CPT and false belief test. Our results suggest that proinflammatory cytokines are associated with cognitive impairment in patients with schizophrenia.

## 1. Introduction

Cognitive impairment is an essential feature of schizophrenia that manifests as difficulty with daily life activities, functional disability at work, and poor social functioning [1]. Cognitive dysfunction is also evident in patients with first-episode schizophrenia and people who have a clinically high risk or a genetic risk for psychosis [2]. Despite recent advances, the pathogenesis of cognitive impairment in patients with schizophrenia is unclear. Two clinical studies have reported that brain inflammation may play a role in neurocognitive functioning in patients with schizophrenia [3,4].

Cognitive impairment has been observed in many psychiatric disorders other than schizophrenia, including major depressive disorder, bipolar disorder, post-traumatic stress disorder, and Alzheimer’s disease, and it remarkably impairs the quality of life of patients and their recovery [5]. Inflammation or the inflammatory response is the result of activation of the immune system, and evidence that psychiatric conditions are accompanied by an activated immune system is growing [6]. Therefore, neuroinflammation has been regarded as a potential underlying cause of cognitive impairment in many brain diseases.

Cytokines, which are small proteins that affect cell function, are essential positive modulators of central nervous system (CNS) functions, including the maintenance of neuroplasticity, which can affect neurocircuit functions such as cognition [7,8]. Cytokines are predominantly produced by immune cells, including microglia in the CNS and other CNS cells in the brain. Immune reactions, including cytokine production, are affected by countless factors, including, but not limited to, genetics and previous exposure to pathogens. Moreover, cytokines can have either pro- or anti-inflammatory effects [9].

Changes in inflammatory cytokines are seen in various psychiatric disorders. High levels of proinflammatory cytokines such as interleukin (IL)-1β, IL-6, and tumor necrosis factor-alpha (TNF-α) have been detected in the brains of patients with Alzheimer’s disease [10]. One meta-analysis suggested that the levels of some cytokines, such as IL-6, IL-12, and TNF-α, are higher in patients with depression [11]. In particular, elderly patients with major depressive disorder have been associated with cognitive decline that may be related to the peripheral IL-6 level [12]. Some studies have described increases in proinflammatory biomarkers such as IL-6 and TNF-α among patients with bipolar disorder. Moreover, activation of peripheral immunity is thought to contribute to cognitive dysfunction in patients with recurring bipolar disorder [13,14].

Changes in immune-inflammatory markers occur in patients with schizophrenia [3]. Inflammatory cytokines have been suggested as potentially useful biomarkers, reflecting the association between inflammation and cognitive function in patients with schizophrenia [15,16]. However, the explainable mechanisms remain largely underexplored. We hypothesized that inflammatory cytokines are associated with cognitive impairment in patients with schizophrenia [1]. We investigated the impact of peripheral inflammatory cytokines on the cognitive performance of patients with recent-onset schizophrenia. To exclude the confounding effects of natural cognitive deterioration, we analyzed the data of patients with first-episode schizophrenia.

## 2. Materials and Methods

### 2.1. Study Design

We analyzed the data of an early psychosis cohort enrolled in the Gwangju Early Treatment and Intervention Team (GETIT) cohort study, which included patients with a recent-onset psychotic disorder [17,18]. The included patients had a duration of psychotic illness of ≤1 year and met the criteria for “Schizophrenia Spectrum Disorder and Other Psychotic Disorders”, according to the Diagnostic and Statistical Manual of Mental Disorders, Fifth Edition [19]. Only patients aged between 15 and 40 years and who had 1 or fewer years of psychotic illness were included to minimize the effects of confounding factors, such as age and duration of illness, on cognitive function and cytokine levels. Patients with a substance- or medication-induced psychotic disorder, psychotic disorder due to another medical condition, or severe neurological or medical disorder were excluded. This study was conducted from January 2015 to December 2020 and was approved by the Chonnam National University Hospital Institutional Review Board (CNUH-2014-225). All subjects provided written informed consent before participation.

### 2.2. Demographic and Clinical Measures

The baseline sociodemographic and clinical data included age, sex, education level, diagnosis, and duration of illness. The dosages of antipsychotics that were prescribed to the patients were converted to the risperidone equivalent dose [20]. The psychiatric measures included the Positive and Negative Syndrome Scale (PANSS) for psychotic symptoms [21,22], the Social and Occupational Functioning Assessment Scale (SOFAS) to evaluate general functioning [23], and the Calgary Depression Scale for Schizophrenia to assess depressive symptoms [24,25].

### 2.3. Neurocognitive Measures

Neurocognitive function was measured by a computerized neurocognitive function test battery, which was standardized within the Korean population, including psychotic patients [26]. Six neurocognitive tests were used in this study including the digit span test, which we administered to assess attention span. Higher scores on the digit span test indicate better attention and vigilance. The verbal learning test (verbal LT), which is the modified Rey auditory verbal LT [27], was administered to assess immediate and delayed verbal memory. The outcome measure was the total number of recalls, in which higher scores indicate better memory. The Wisconsin card sorting test (WCST) was administered to assess executive function [28]. The outcome measure included the number of categories completed, in which a higher score indicates better executive function and cognitive flexibility. The continuous performance test (CPT) was administered to measure the ability to sustain attention (vigilance) to a stimulus for a period of time [29]. The outcome measures included reaction time and correct responses. A shorter reaction time and more correct responses indicate better sustained attention. The visual learning test (visual LT) was administered to assess immediate and delayed visual memory. The outcome measure was the total number of recalls, and a higher score indicated better memory [30]. The trail making test (TMT) was administered to assess visuomotor coordination speed based on a timed visual tracking task [31]. The TMT consists of two parts (A and B) that measure executive function and visuospatial working memory. The outcome measures were the scores on parts A and B, obtained by recording the time (in seconds) to complete each task. Higher scores indicate poorer performance [32].

### 2.4. Social Cognitive Measures

The false belief task of Wimmer and Perner was administered to assess social cognition. Four stories were provided, and the patients were told to infer the thoughts of the characters in the story. The total score was 12 points, and the higher the score, the better the ability of the theory of mind (ToM) [33]. The picture story task developed by Brüne was administered to assess the ToM, which is the ability to infer the mental states and emotions of others [34]. Six cartoon tests were presented to the patients to evaluate their answers to questions related to ordering and mind-reading. The total scores for the ordering (total of 36 points) and mind-reading questions (total of 23 points) were analyzed. The higher the score, the better the ToM ability [35].

### 2.5. Cytokines Measurements

A venous blood sample was collected on the same day as the baseline clinical assessments. All samples were processed within 2 h of blood collection and stored at −80 °C until further analysis. We quantitatively determined the steady-state levels of the circulating inflammatory cytokines of interest, including TNF-α, interferon (IFN)-γ, IL-1β, IL-6, IL-8, IL-10, and IL-12. Plasma concentrations were determined using a human cytokine/chemokine magnetic bead panel (Milliplex MAP Kit; Millipore Corp., Billerica, MA, USA) according to the manufacturer’s instructions. A Luminex Bio-Plex 100 Analyzer (Magpix; Luminex Corp., Austin, TX, USA) was used to identify individual microspheres, and the results were quantified based on fluorescent reporter signals using Milliplex Analyst 5.1 (Merck KGaA, Darmstadt, Germany) and Luminex xPONENT (Luminex Corp.) acquisition software. We analyzed the median fluorescence intensity using a five-parameter logistic or spline curve-fitting method to calculate the cytokine concentrations in each sample. All assays and quality control were performed by a laboratory company (GC Cell, Yongin, Korea) according to the manufacturer’s instructions.

### 2.6. Statistical Analysis

Spearman’s correlation analysis was performed to assess the associations between cytokine levels and cognitive and clinical measures. Differences in cytokine levels and/or clinical measures according to sex and type of antipsychotic were analyzed with the Mann–Whitney U test or Kruskal–Wallis test. When a significant association was observed between the level of a cytokine and a clinical measure, partial correlation analysis was conducted after log transformation of the cytokine level to adjust for confounding effects. All statistical tests were two-tailed, and a *p*-value < 0.05 was considered significant. The statistical analysis was performed using SPSS software (version 25.0; IBM Corp., Armonk, NY, USA).

## 3. Results

A total of 174 patients (44.8% men and 55.2% women) with first-episode psychosis were enrolled in this study. The median (interquartile range) age at baseline was 24 (20–29) years. The median (interquartile range) duration of psychotic illness and education were 3 (2–6) months and 14 (12–16) years, respectively. The most common diagnosis of the participants was schizophrenia (*n* = 111, 63.8%), followed by schizophreniform disorder (*n* = 47, 27.0%), other specified schizophrenia spectrum disorder (*n* = 10, 5.7%), and schizoaffective disorder (*n* = 6, 3.4%). The mean ± standard deviation scores on the PANSS and SOFAS were 65.8 ± 13.1 and 59.8 ± 8.2, respectively.

Table 1 shows Spearman’s correlation coefficients between the cognitive measures and clinical measures. Age significantly positively correlated with CPT reaction time but negatively correlated with correct responses on the CPT. Education positively correlated with the ToM test, backward digit span test, verbal LT, and visual LT. The PANSS negative subscale significantly positively correlated with TMT-type B and negatively correlated with the visual LT. The SOFAS score significantly negatively correlated with TMT-type A and B times. Cognitive measures did not significantly differ according to sex, except for the verbal and visual LTs; the scores for these measures were significantly higher for female than male patients (*p*-values < 0.001, data not shown).

Spearman’s correlation coefficients between the cytokines and clinical measures are shown in Table 2. The cytokines were not initially associated with most of the demographic and clinical measures, including the dosage of antipsychotics, education, or the PANSS total score. A negative correlation was observed between the PANSS positive subscale score and IL-12, and between age and IL-6 level. Appendix A shows comparisons of cytokine levels according to the type of antipsychotic and sex. Cytokine levels did not significantly differ according to the type of antipsychotics used (all *p*-values > 0.05). The IFN-γ, IL-1β, and IL-12 levels were significantly higher in female than male patients (Appendix A).

Table 3 shows the correlation coefficients between cytokines and cognitive measures. Partial correlation coefficients were calculated in the analyses of IFN-γ, IL-1β, IL-6, and IL-12 levels, after adjusting for clinical measures that were significantly associated with these cytokines. The TNF-α level negatively correlated with the scores on the forward digit span test, verbal LT, and WCST, and with the number of correct responses on the CPT, but positively correlated with the TMT-type B time. The IFN-γ level negatively correlated with performance on the false belief test and visual LT after adjusting for sex. The IL-1β level positively correlated with the CPT reaction time and TMT-type A time, but negatively correlated with scores on the false belief and ToM tests, and with the number of correct responses on the CPT and visual LT after adjusting for sex. The IL-12 level negatively correlated with the number of correct responses on the CPT and scores on the false belief test after adjusting for sex and the PANSS-positive subfactor score. The IL-6, IL-8, and IL-10 levels were not correlated with any cognitive measures.

## 4. Discussion

An activated immune system, as measured by increases in circulating cytokines, was associated with cognitive impairment in patients with schizophrenia [36]. In particular, proinflammatory cytokine levels increase in the blood [37] and may affect cognitive functioning in patients with schizophrenia [4]. In this study, the correlations between peripheral inflammatory cytokine levels and cognitive impairment were investigated in patients with first-episode psychosis. Our findings showed that the levels of proinflammatory cytokines TNF-α, IFN-γ, IL-1β, and IL-12 correlated with cognitive dysfunction.

Microglia are critical regulators of inflammation in the brain [38] that modulate brain functions, including synapse modification and the production of proinflammatory cytokines such as TNF-α, IL-6, and IL-1β, which provoke the inflammatory response and may disrupt the blood–brain barrier (BBB) [39]. Inflammatory cytokines are critical in synaptogenesis and neurodegeneration, suggesting a possible mechanism for the link with neurocognition [40]. Furthermore, poor cognitive functioning and disability in daily life activities are negatively associated with increased peripheral TNF-α and IL-12p70 levels in patients with schizophrenia [41]. These findings suggest that neuroinflammation is a cause of cognitive impairment in patients with recent-onset schizophrenia.

In our study, the TNF-α level negatively correlated with various cognitive functions, including attention span, verbal memory, executive function, sustained attention, and psychomotor speed. These findings are consistent with those of a previous study showing a positive association between cognitive dysfunction and TNF-α levels in patients with chronic schizophrenia [42]. TNF-α is crucial in brain development, modulating synaptic pruning and plasticity [43]. TNF-α produced by microglia appears to act in cognitive dysfunction in patients with schizophrenia [44]. Studies with rats illustrated the negative effects of TNF-α on cognitive function, including decreased long-term potentiation, a correlate of learning and memory [45]. Moreover, the positive effects on depressed patients who used an antibody against TNF-α support the concept that brain function may be affected by inflammatory cytokines that are modestly correlated with cognitive function [46].

IL-1β is important in the cognition of patients with schizophrenia, as it is needed for long-term potentiation of synaptic transmission and seems to play a crucial role in hippocampal neurogenesis [47,48]. The hippocampus appears to be highly vulnerable to the effects of neuroinflammation and proinflammatory cytokines [49], potentially leading to impaired memory and learning [50]. In this study, the IL-1β level negatively correlated with scores on visual learning and sustained attention tasks.

IL-1β, IL-12, and IFN-γ levels negatively correlated with the ToM. Social cognition is one of the best predictors of social functioning in patients with schizophrenia [51]. A previous randomized controlled trial showed that injecting endotoxin leads to inflammation with a significant increase in proinflammatory cytokine levels (e.g., IL-6 and TNF-α) and impaired ability to perceive and correctly infer the emotional states of others [52]. These results demonstrate that experimental inflammation may lead to deficits in social cognition. This observation agrees with our data, as proinflammatory cytokines negatively correlated with the ToM test. Relevant evidence suggests that abnormal social cognition in patients with schizophrenia may reflect stress-induced neuroinflammatory degeneration of the neural circuitry in the brain that regulates the ToM [53]. The social-cognitive process activates the innate immune response. This process upregulates genes for pro- and anti-inflammatory cytokines [54,55].

IL-6 is a pleiotropic cytokine with pro- and anti-inflammatory properties. Serum IL-6 levels are significantly higher in patients with schizophrenia, and there is a negative correlation between IL-6 and cognition [56]. A recent study suggested that IL-6 regulates cognitive function in patients with schizophrenia via its action in the choroid plexus and BBB [57]. In this study, IL-6 was not significantly associated with the cognitive measures after adjusting for age as a covariate.

A few studies have reported associations between proinflammatory cytokine levels and acute psychotic symptoms [39]. However, meta-analyses have demonstrated that most cytokines (IL-12, IL-10, IFN-γ, TNF-α, and the IL-2 receptor) appear to be trait makers, as their levels are unchanged following antipsychotic treatment [58]. One meta-analysis reported that proinflammatory cytokine levels decrease after antipsychotic treatment [59], whereas another study found no differences in cytokine levels following antipsychotic treatment in a group of stable, medicated outpatients with schizophrenia [60]. In the present study, the association between cytokine levels and psychotic symptoms was weak compared with the association between cytokine levels and cognitive function. This might be attributable to the characteristics of the population, who were generally symptomatically stable and on antipsychotic medication. In addition, the type, dosage, and duration of antipsychotic treatment were not associated with the cytokine levels. Our findings suggest that the association between proinflammatory cytokine levels and cognitive impairment may be a potential trait marker in patients with schizophrenia.

A previous study researching the N-methyl-D-aspartate (NMDA) receptor antagonist MK-801 in mice suggested that β-asarone improves cognitive function, including memory and social deficits [61]. Long-term potentiation is a key mechanism underlying the ability of the hippocampus to store information that can be eliminated by NMDA receptor antagonists [62]. β-asarone mediates microglial activation and significantly downregulates the levels of proinflammatory cytokines such as IL-6 and IL-1β, which were markedly increased in the hippocampus. This also suggests that hippocampal inflammatory cytokines are involved in cognitive deficits [61].

Previous studies showed that the anti-inflammatory cytokine IL-10 is neuroprotective and prevents neuronal dysfunction [63]. IL-10 is positively correlated with the ToM in chronic patients with delusions but not in nondelusional patients or healthy controls [53]. However, our study did not show that anti-inflammatory cytokines were not positively correlated with cognitive function. Few studies have linked the role of IL-10 with cognitive deficits in schizophrenia [64]. Moreover, the dualism of pro- and anti-inflammatory cytokines is a simplified concept because some cytokines have both pro- and anti-inflammatory properties, depending on several factors [65]. IL-10 maintains a balance between pro- and anti-inflammatory cytokine levels in microglia by inhibiting glial activation [66]. Data regarding the levels of IL-10 in patients with schizophrenia are mixed; they have been reported to be higher and lower in first-episode patients than in controls [58,67,68]. IL-10 has not been associated with the neurocognitive domains of verbal memory or executive functioning (trials A and B) [69,70].

An anti-inflammatory treatment trial on cognitive function can be considered based on our study results. Several randomized clinical trials have been conducted on nonsteroidal anti-inflammatory drugs as adjuncts to antipsychotics [71], and other trials that investigated adjunctive treatments to antipsychotics included minocycline, a second-generation tetracycline with an anti-inflammatory effect. A double-blind, randomized, controlled trial using minocycline augmented with an antipsychotic medication reported improved executive functioning compared to controls [72]. Quantitative evidence through a recent meta-analysis demonstrated that visual learning/memory, attention, and executive function significantly improved after minocycline augmentation [73].

N-acetylcysteine (NAC), a glutathione precursor with potent antioxidant, proneurogenesis, and anti-inflammatory properties [74], appears to ameliorate cognitive dysfunction in patients with psychiatric disorders linked to dysregulation of NMDA [75]. Another analysis of patients with bipolar disorder and schizophrenia using NAC revealed significant improvement in working memory function [76]. A double-blind, randomized, placebo-controlled trial of early psychosis patients reported significant improvement in neurocognition by NAC (processing speed) [77]. An open-label trial for the IL-6 receptor monoclonal antibody tocilizumab in patients with schizophrenia resulted in cognitive improvement, although the improvement was not replicated in later studies [78,79]. Thus, further study is needed. Cognitive dysfunction in all patients with schizophrenia may not be attributed to changes in inflammation. However, our study adds to a growing body of work, and suggests the need for further study to investigate the effects of medications targeting a specific cytokine associated with cognitive function in patients with schizophrenia. This approach may contribute to improving cognitive functioning in patients with schizophrenia, which has not been effectively treated. In terms of precision medicine, we may consider the potential benefit of anti-inflammatory drugs that can improve cognitive function in patients with schizophrenia who have elevated cytokine levels.

This study had several limitations. First, we only analyzed the cytokine levels at baseline and therefore lacked data for longitudinal analysis. The cross-sectional design of the study prevented any inferences being drawn regarding causal relationships. Second, we measured plasma cytokines but not those in cerebrospinal fluid. It is still uncertain whether peripheral cytokines reflect similar changes in the brain. Third, few patients were in a drug-naïve state. Although we found no significant effects of antipsychotic medication or the duration of illness on cognitive functioning or cytokine levels, assessing patients in a drug-naïve state would be optimal. Finally, potential type I error should be considered because we obtained empirical *p*-values rather than correcting for multiple testing. Nevertheless, our results provide a basis for further studies to investigate the role of specific proinflammatory cytokines and novel treatment strategies.

## 5. Conclusions

Our findings indicate that higher proinflammatory cytokine levels in patients with first-onset schizophrenia may be associated with cognitive impairment in the domains of visual and verbal memory, psychomotor speed, sustained attention, and social cognition. This finding contributes to future studies exploring mechanisms of cognitive impairment in patients with schizophrenia, and provide a basis for precision medicine to effectively treat cognitive impairment in young patients with recent-onset schizophrenia.

## Figures and Tables

**Table 1 jpm-12-01137-t001:** Spearman correlation coefficients between cognitive measures and clinical measures.

	Age	Education	PANSS_P	PANSS_N	PANSS_G	PANSS_T	SOFAS	CDSS
**False Belief Test**	0.005	0.065	−0.123	−0.088	−0.111	−0.130	0.025	−0.127
**Theory of Mind**	0.027	0.177 *	−0.095	−0.136	−0.092	−0.135	0.036	0.008
**DS, forward**	−0.032	0.080	0.071	−0.095	0.016	0.001	0.046	0.014
**DS, backward**	0.093	0.208 **	0.090	−0.083	0.016	0.022	−0.017	−0.027
**Verbal Learning Test**	−0.059	0.160 *	−0.046	−0.110	−0.061	−0.098	0.088	−0.013
**WCST_CC**	−0.085	0.115	0.009	−0.135	−0.030	−0.066	0.103	−0.037
**CPT-CR**	−0.175 *	−0.082	0.081	−0.043	0.019	0.026	0.064	0.079
**CPT-RT**	0.343 ***	0.133	−0.081	−0.099	−0.104	−0.123	0.018	−0.027
**Visual Learning Test**	−0.083	0.182 *	0.050	−0.189 *	0.016	−0.036	0.126	0.038
**TMT-A**	−0.032	−0.120	−0.041	0.188	0.138	0.118	−0.196 **	0.070
**TMT-B**	0.073	−0.103	0.034	0.218 **	0.121	0.147	−0.202 **	0.063

* *p*-value < 0.05, ** *p*-value < 0.01, *** *p*-value < 0.001. PANSS, Positive and Negative Syndrome Scale; PANSS_P, PANSS_Positive; PANSS_N, PANSS_Negative; PANSS_G, PANSS_General; PANSS_T, PANSS_Total; SOFAS, Social Occupational Functioning Assessment Scale; CDSS, Calgary Depression Scale for Schizophrenia; DS, digit span test; WCST_CC, Wisconsin card sorting test_category completed; CPT, continuous performance test; CPT-CR, CPT-correct responses; CPT-RT, CPT-reaction time; TMT-A, trail making test type A time; TMT-B, TMT-type B time.

**Table 2 jpm-12-01137-t002:** Spearman correlation coefficients between cytokines and clinical measures.

	TNF-α	IFN-γ	IL-1β	IL-6	IL-8	IL-10	IL-12
**PANSS_P**	0.018	−0.122	−0.098	0.096	0.041	0.081	−0.156 *
**PANSS_N**	0.076	−0.052	0.051	0.065	0.076	−0.022	0.002
**PANSS_G**	0.080	−0.069	−0.052	0.050	0.022	−0.016	−0.102
**PANSS_T**	0.080	−0.097	−0.047	0.072	0.050	0.002	−0.112
**SOFAS**	−0.106	0.032	−0.001	−0.065	−0.096	−0.013	−0.003
**CDSS**	0.009	0.061	0.016	−0.034	−0.068	0.000	−0.040
**Age**	0.003	−0.037	−0.039	−0.204 **	−0.076	−0.082	−0.076
**Education**	−0.031	−0.030	−0.011	−0.028	−0.026	0.083	−0.001
**Antipsychotic dosage**	0.126	−0.002	0.072	0.077	0.089	−0.019	−0.009
**Treatment duration**	−0.090	−0.108	−0.091	−0.113	0.039	−0.121	−0.089

All *p*-values are empirical. * *p*-value < 0.05, ** *p*-value < 0.01. TNF-α, tumor necrosis factor-alpha; IFN-γ, interferon-gamma; IL, interleukin; PANSS, Positive and Negative Syndrome Scale; PANSS_P, PANSS_Positive; PANSS_N, PANSS_Negative; PANSS_G, PANSS_General; PANSS_T, PANSS_Total; SOFAS, Social Occupational Functioning Assessment Scale; CDSS, Calgary Depression Scale for Schizophrenia.

**Table 3 jpm-12-01137-t003:** Correlation coefficients between cytokines and cognitive measures.

	TNF-α ^a^	IFN-γ ^b^	IL-1β ^b^	IL-6 ^c^	IL-8 ^a^	IL-10 ^a^	IL-12 ^d^
**False Belief Test**	−0.096	−0.195 *	−0.184 *	−0.097	−0.050	−0.140	−0.203 **
**Theory of Mind**	−0.075	−0.109	−0.163 *	−0.073	−0.034	−0.091	−0.138
**DS, forward**	−0.161 *	−0.042	−0.061	0.053	0.037	0.047	0.013
**DS, backward**	−0.072	0.062	0.000	−0.013	0.074	0.149	0.088
**Verbal Learning Test**	−0.184 *	−0.074	−0.127	−0.084	−0.136	−0.003	−0.033
**WCST_CC**	−0.182 *	−0.090	−0.057	0.046	−0.014	−0.026	−0.077
**CPT-CR**	−0.205 **	−0.130	−0.174 *	−0.107	−0.057	−0.035	−0.166 *
**CPT-RT**	0.082	0.146	0.161 *	0.065	−0.051	−0.008	0.065
**Visual Learning Test**	−0.090	−0.168 *	−0.191 *	−0.012	0.004	0.033	−0.117
**TMT-A**	0.089	0.105	0.188 *	−0.022	−0.125	−0.028	0.081
**TMT-B**	0.171 *	0.016	0.103	0.059	−0.099	0.019	0.090

All *p*-values are empirical. * *p*-value < 0.05, ** *p*-value < 0.01. ^a^ correlation coefficient calculated by Spearman’s correlation analysis; ^b^ correlation coefficient adjusted for sex (partial correlation); ^c^ correlation coefficient adjusted for age (partial correlation); ^d^ correlation coefficient adjusted for sex and the Positive and Negative Syndrome Scale positive subscale score (partial correlation). TNF-α, tumor necrosis factor-alpha; IFN-γ, interferon-gamma; IL, interleukin; DS, digit span test; WCST_CC, Wisconsin card sorting test category completed; CPT, continuous performance test; CPT-CR, CPT-correct responses; CPT-RT, CPT-reaction time; TMT-A, trail making test type A time; TMT-B, TMT-type B time.

## Data Availability

The data presented in this study are available upon reasonable request to the corresponding author.

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
