# Peer review of "Association between Peripheral Inflammatory Cytokines and Cognitive Function in Patients with First-Episode Schizophrenia"

_jpm, 2022, doi:10.3390/jpm12071137_

Round 1
Reviewer 1 Report
This is a well-written original paper on an interesting topic. The notion of immune activation and neuroinflammation as a crucial component of schizophrenia's pathophysiology merits extensive research and the present paper adds significant information.
However the cross-sectional methodological design prevents us from drawing safe conclusions regarding the direction of the observed associations. In addition, the authors should provide more specific information regarding the type of antipsychotic medication given that certain drus may have inflammation-modulatory properties that might affect outcomes. For example it would be interesting to know whether the use of specific antipsychotics (e.g. clozapine) was associated with a different cytokine profile compared to the other drugs.
Furthermore, the authors are encouraged to further comment on the finding that PANSS positive scores were strongly correlated with IL12, however no other significant association emerged between clinical indices of disease severity and cytokine levels. In my opinion it is quite interesting that cytokine levels were correlated with cognitive function but not with schizophrenia symptoms and this should be explained in the discussion section. An alternative explanation might be that antipsychotic medication and not schizophrenia mediates the inflammation-associated cognitive decline. For this reason, the authors should provide data on the association between schizophrenia symptomatology, functional impairment, medication use and cognitive functioning.
Reviewer 2 Report
Inflammation and cytokines are important in schizophrenia and, as the authors say, may lead to new therapies.
I would like to see a statistical correction for multiple testing.
I am also curious why 55.2% of first episode schizophrenia patients with an average age of 24 were women. In North America and Europe most young first episode schizophrenia patients would be men. Is there an explanation for the relative percentages of men and women and were there cognitive differences between men and women?
Round 2
Reviewer 1 Report
The paper may be accepted for publication in its revised form